# The Optimal Explorer Hypothesis and Its Formulation as a Combinatorial Optimization Problem

**Mikel Malagón** [*]
University of the Basque Country UPV/EHU
Donostia-San Sebastián
mikel.malagon@ehu.eus

**Jon Vadillo**
University of the Basque Country UPV/EHU
Donostia-San Sebastián
jon.vadillo@ehu.eus

**Josu Ceberio**
University of the Basque Country UPV/EHU
Donostia-San Sebastián
josu.ceberio@ehu.eus

**Jose A. Lozano**
Basque Center for Applied Mathematics
University of the Basque Country UPV/EHU
Donostia-San Sebastián
ja.lozano@ehu.eus

## Abstract

This research project explores the hypothesis that, given a bounded number of steps in an environment, agents that most efficiently optimize their model of the environment are more likely to induce emergent intelligent behavior in a reward-free scenario. We refer to this as the optimal explorer hypothesis. The project aims to formalize and analyze this hypothesis, investigating its theoretical implications and connections to related areas such as open-ended learning and active inference. Building on this foundation, we will develop a practical implementation of an approximate "optimal explorer" agent by formulating it as a combinatorial optimization problem and leveraging established methods from the field. Finally, we will conduct extensive experiments to evaluate whether the proposed agent induces emergent behaviors in diverse and challenging environments.

## 1   Introduction

Agents—understood as systems acting by themselves according to certain goals or norms in an environment [Barandiaran et al., 2009]—are the substrates of intelligent life as we know it. However, not all agents can be classified as intelligent. For instance, a thermostat fits into most definitions of agency, while not exhibiting intelligence as found in biological agents. This raises the question: *what are intelligent agents doing?* In other words, what are the objectives that intelligent agents pursue that give rise to the incredibly complex emergent behaviors that we broadly observe in natural life?

On the Artificial Intelligence (AI) field side, Reinforcement Learning (RL) has been the area of research that has most prominently focused on the subject of intelligent agents [Kaelbling et al., 1996, Sutton and Barto, 2018]. Although RL has led to many breakthroughs in the last decades [Silver et al., 2016, Abramson et al., 2024], most RL literature has focused on developing agents to pursue a single, well-defined objective. In fact, Sutton's *reward hypothesis* states that all goals can be framed as cumulative reward maximization [Sutton, 2004, Bowling et al., 2023]. This led Silver et al. [2021] to hypothesize that optimizing reward can lead to the emergence of general intelligence and complex behavior in a sufficiently rich environment Silver et al. [2021].

On the other hand, as discussed by Stanley and Lehman [2015] and Soros et al. [2017], following an explicit objective can lead to dead ends. These works challenge the effectiveness of explicit

---

[*]Corresponding author.

XVI XVI Congreso Español de Metaheurísticas, Algoritmos Evolutivos y Bioinspirados (maeb 2025).

objectives, arguing that direct goal optimization often fails to discover necessary stepping stones. It emphasizes open-ended exploration over direct optimization, suggesting that breakthroughs arise from serendipity and novelty search rather than predefined goals [Lehman and Stanley, 2011, Kumar et al., 2024]. The emergence of intelligent behavior via open-ended novelty search has inspired a growing number of works in recent years [Bauer et al., 2023, Bruce et al., 2024, Matthews et al., 2025], even characterizing it as essential for superhuman-level intelligence [Hughes et al., 2024]. Many of these works focus on automating an open-ended environment (task) generation process [Bruce et al., 2024, Faldor et al., 2025] and learn a robust policy that will generalize to unseen tasks, known as Unsupervised Environment Design (UED) [Parker-Holder et al., 2022, Bauer et al., 2023, Rigter et al., 2024, Beukman et al., 2024]. However, most open-ended literature assumes that the learning method has access to and control of the environment to generate vast amounts of tasks (e.g., UED) or high-level control (e.g., Voyager by Wang et al. [2024]).

Instead, in natural life, agents interact with a (single) complex environment only through perception and (low-level) action. Since Helmholtz [1867], most prominent theories of cognition of today agree that the brain maintains and updates a model of its environment (i.e., the real world) [Doya, 2002, Friston, 2009]. Based on these ideas and recent work on lifelong learning and open-endedness theory [Abel et al., 2023, Hughes et al., 2024] this work hypothesizes that (informally):

> *The agents that most efficiently learn an internal model of the environment are more likely to produce emergent intelligent behavior in reward-free scenarios over a bounded time scope.*

In this context, the agent's model of the environment—referred to as the *world model*—is trained on agent-generated trajectories (i.e., sequences of interactions with the environment). Efficiency is measured as the expected sum across timesteps of the world model's prediction error with respect to the environment over all the possible trajectories. We refer to this as the *optimal explorer hypothesis*. Note that this hypothesis does not state that the agents that most efficiently learn their world model are the only or the most likely ones to induce emergent behaviors, just that they are more likely to cause them by doing so.

In the search for emergent behavior, this hypothesis directly introduces the intrinsic objective of acting to generate the most informative trajectories for the world model in the long run. Based on this hypothesis and literature on active inference [Friston, 2009] and model-based RL [Chua et al., 2018], the next part of this project will propose a practical implementation of an agent to optimize this long-term intrinsic objective. Equipped with a deep Neural Network (NN) ensemble-based world model [Lakshminarayanan et al., 2017], we aim to introduce an agent that plans and selects the sequences of actions that maximize the world model's epistemic uncertainty, in the long run, using the Cross-Entropy Method (CEM) [Rubinstein, 1999]. This way, the action selection policy and the world model (constantly updated with the trajectories sampled by the latter) play a minimax game that explores in face of the unknown while otherwise exploiting to explore. Finally, we will conduct an extensive empirical evaluation on challenging environments to analyze the behavior of the proposed agent. We expect the agent to solve complex games (in episodic setups) even without having a reward signal, and to improve the sample efficiency of reward-based RL methods model-based (e.g., DreamerV3 [Hafner et al., 2023]) and non-model-based methods (e.g., proximal policy optimization [Schulman et al., 2017]).

In summary, the main objectives of this project are the following:

1. **Formalization of the optimal explorer hypothesis.** Define and analyze the hypothesis, establishing its theoretical foundations and connections to related research areas such as open-ended learning and active inference.

2. **Combinatorial optimization formulation.** Frame the problem of optimal exploration as a Combinatorial Optimization (CO) task, identifying suitable problem representations and constraints.

3. **Algorithm development.** Design and implement an approximate optimal explorer agent by leveraging techniques from model-based RL and combinatorial optimization, such as Estimation of Distribution Algorithms [Larrañaga and Lozano, 2002] (employed in the CEM).

4. **Empirical evaluation.** Conduct experiments in diverse and challenging environments to assess the effectiveness of the proposed agent in inducing emergent behaviors in reward-free scenarios.

## 2   Previous work

The following lines provide a brief overview of the fields and work upon which this work is mainly based.

**Lifelong and open-ended learning.**   Lifelong and open-ended learning focus on agents that continuously acquire and refine knowledge over time, adapting to novel scenarios by leveraging past experiences. However, learning continuously introduces many challenges as catastrophic forgetting and interference, loss of plasticity, or computational cost [Hadsell et al., 2020]. Addressing these issues is an active area of research [Khetarpal et al., 2022, Wolczyk et al., 2024, Malagon et al., 2024]. Other works depart from sequential tasks and focus on meta-learning a robust policy on a distribution of environments [Parker-Holder et al., 2022, Beukman et al., 2024]. Although these deeply connected fields have gained increasing attention in recent years, they are still in the phase of formally defining themselves [Abel et al., 2023, Hughes et al., 2024].

**Exploration strategies.**   Although many goals can be framed as a reward maximization problem [Sutton, 2004], learning a policy can be extremely difficult in the absence of a dense informative reward signal. Thus, the field of RL has come with a vast body of work on intrinsic reward: an auxiliary reward function to guide exploration to promising trajectories [Pathak et al., 2017, Burda et al., 2019, Nikulin et al., 2023]. Even with intrinsic motivation, RL agents greatly suffer from sample efficiency. In this realm, model-based methods learn (or directly employ when available) a model of the environment which is used to plan the actions [Kaiser et al., 2020, Hafner et al., 2023]. However, model-based RL incorporates additional complexity and agents can exploit biases in the model that lead to substantial degradation of performance in these types of methods [Janner et al., 2019].

**Active inference.**   Active inference is based on Friston's Free Energy Principle (FEP) [Friston, 2009]. According to the FEP, living beings minimize expected free energy, maximizing the probability of being in desirable states (maintaining homoeostatic equilibrium) while maximizing information gain (minimizing epistemic uncertainty) in the long run [Friston et al., 2015]. Despite the appeal of biologically plausible active inference agents [Friston, 2010] and recent efforts to incorporate deep neural networks [Fountas et al., 2020], scaling beyond toy environments remains a challenge for these methods [Sajid et al., 2021].

## 3   The optimal explorer hypotheis

As described in the introduction, we focus on agents that maintain and update an internal model (i.e., world model) of their environment.[2] Moreover, the environment is only composed of a transition function and without a reward function (i.e., reward-free environments). Every timestep the agent interacts with the environment by generating a new transition, and the world model is updated accordingly.

In this setup, we hypothesize that the agents that generate the trajectories (sequences of interactions) that most efficiently update their world model are more likely to induce emergent intelligent behavior in a finite scope of time. In this context, we define the efficiency of an agent as the expected sum of the global world model error at each timestep by following the agent's policy.[3] In turn, we refer to global error as the world model's error modeling of the environment given all the possible trajectories. Thus, if the global error is zero, the world model and the environment define the same probability distribution.

---

[2]The world model can be naturally defined as the distribution over all the possible states given the current state and action, $p_\phi(s_{t+1}|s_t, a_t)$.

[3]We refer to an agent's policy in the classic RL sense, that is, the probability distribution over actions given the current state $p(a|s)$.

Intuitively, those agents that most efficiently optimize their world models will be those that find (often by exploiting *shorcuts* in complex environments) the best trajectories to explore their environments. Note that this substantially differs from random exploration (e.g., $\epsilon$-greedy exploration), as the most efficient agents will be those that exploit to explore. For instance, in an episodic environment such as an Atari game [Machado et al., 2018] (and most games) an efficiently exploring agent (in terms of our hypothesis) would have to solve the game as fast as possible to update its model with interactions from advanced stages of the game.

## 4 Proposing a practical implementation based on CO

In this part of the project, we aim to explore the implications of the hypothesis proposed in the previous section. Specifically, we leverage the ideas from the optimal explorer hypothesis to propose an agent that efficiently explores its environment in the absence of a reward function (i.e., without explicit objectives). Note that many possible implementations of such an agent exist and that the one from this part of the project is just a proposal to analyze the experimental implications of the hypothesis.

Specifically, we aim to leverage previous work on uncertainty quantification for deep NN models [Gal and Ghahramani, 2016, Lakshminarayanan et al., 2017] to select those trajectories that are more informative to the world model (a deep NN). Finding the action that will cause the most efficient world model update (in terms of the hypothesis) at each timestep can be framed as searching for the action sequence that will lead the agent toward the most informative interactions—of highest epistemic uncertainty—for the world model.

Note that this problem can be framed as a **combinatorial optimization** problem where the space of **possible solutions** $\Omega$ are the sequences of actions, $a \in \mathcal{A}$, $\mathbf{x} = (a_1, a_2, \ldots, a_n)$ of a given length $n$ (that corresponds to the planning horizon).[4] Accordingly, the **objective function** $f(x)$ is the cumulative epistemic uncertainty of the world model at each interaction, where the interactions are autoregressively sampled from the world model itself. Formally, considering the world model a probability distribution parametrized by $\phi$ over the next states of the environment conditioned on the current action and state, the objective function $f$ can be written as,

$$f(\mathbf{x}, i, s) = EU(p_\phi(\cdot|s, \mathbf{x}_i)) + \mathbb{E}_{s' \sim p_\phi(\cdot|s, \mathbf{x}_i)}[f(\mathbf{x}, i+1, s')]. \tag{1}$$

Where $EU(p_\phi(\cdot|s, \mathbf{x}_i))$ is the epistemic uncertainty of the state $s$ and action $\mathbf{x}$ in the world model $p_\phi$. Note that the fitness of a solution $\mathbf{x}$ is always given with respect to a specific state $s$, as the utility of a given action sequence is completely dependent on the initial state in which it is taken. Thus, our agent proposal, being in a state $s$, would select the action $a$ such that,

$$a = \underset{a_1 \in \mathcal{A}}{\arg\max} f((a_1, \ldots), 1, s). \tag{2}$$

Intuitively, at each state, the agent would choose the action that maximizes the expected long-term epistemic uncertainty of its world model.

## 5 Conclusion

This project outlines a novel approach to emergent intelligent behavior in the absence of explicit objectives (i.e., without reward function). We first (informally) introduce the optimal explorer hypothesis, which connects the efficiency of learning a model of the environment and the likelihood of inducing emergent behaviors. From this hypothesis, we structure the project into four objectives: (1) formalizing the hypothesis, (2) formulating it as a combinatorial optimization problem, (3) developing an agent based on the combinatorial optimization problem formulation, and (4) extensive empirical analysis of the agent. The project aims to advance our comprehension of exploration strategies and learning dynamics in artificial agents, paving the way for more adaptable and intelligent systems and their formal understanding.

---

[4]Where $\mathcal{A}$ is finite and its elements discrete.

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
