# OpenReview forum: "The Optimal Explorer Hypothesis and Its Formulation as a Combinatorial Optimization Problem"
_MAEB/2025/Projects_Track — MAEB 2025 Proyectos_

### Official Review · Reviewer_AzMW · 2025-03-17
**Tentative review for The Optimal Explorer Hypothesis and Its Formulation as a COP**

**Rating:** 3
**Confidence:** 2

**Review:**

The paper proposes testing the hypothesis of the optimal explorer as a formulation of a combinatorial problem.

The authors provide an extensive list of references on the subject, the writing is correct and well-organized, and the approach appears coherent. However, for an inexperienced reader in this field (me), the following doubts arise:

- Are the definitions of efficiency and intelligent behavior correct? I have understood that the emergence of intelligent behavior is being defined using a fitness function that directly depends on the reduction of uncertainty in the environment where the agent operates. This function seems to be entirely correlated with the definition of efficiency as the agent's ability to reduce the error between the environment and its internal world model. That is, the hypothesis could appear to be reduced to: "reducing uncertainty reduces the error in our predictions about the environment." Therefore, with these definitions, isn't the optimal explorer hypothesis a triviality?

- On the other hand, I notice that the objective function depends on a planning horizon, which reminds me of the No-Free-Lunch theorems for optimization. Very briefly, these theorems state that no perfect strategy exists for tackling a problem when no prior knowledge is available, since the characteristics of the problem could be such that the planned strategy turns out to be exactly the worst possible. This raises the question: Is any mathematically possible environment being considered here? If so, I fear that no strategy or method for generating the action planning proposed in the paper would be generally useful (except for a strategy that simply avoids re-evaluating known states => in most cases, random search).

---

### Official Review · Reviewer_nTeS · 2025-03-19
**Review for "The Optimal Explorer Hypothesis and Its Formulation as a Combinatorial Optimization Problem"**

**Rating:** 5
**Confidence:** 4

**Review:**

Intelligent agents keep and update world models internally as they choose actions that affect the environment changing its state. This work plans to analyze the optimal explorer hypothesis, which claims that the agents that most efficiently update the world model are the ones that most likely induce emergent intelligent behaviour in a given bounded time.

The plan of this project is to formulate this hypothesis as a combinatorial optimization problem and solve it. In particular, the authors use a deep neural network as a world model and evaluate the epistemic uncertainty of the world model (neural network) given a state and action. The solutions are the sequences of action chosen by the agent in a given timeframe and the objective function is a recursive formula including the epistemic uncertainty.

I found the idea interesting and I think it should be explored. Definitly, in my opinion, it should be accepted to present the result in MAEB.

I have, however, a doubt regarding how the hypothesis will be checked. There are two parts in the hypothesis:

1. the agents that most efficiently update the world model ...
2. are the ones that most likely induce emergent intelligent behaviour in a given bounded time.

I assume that the combinatorial optimization problem aims at building an agent satisfaying part 1. However, how to measure the emergent intelligent behaviour is not clear to me. I think this should have been better clarified in the paper.

I also found a couple of typos:
- Line 149: "are autoregressive[ly] sampled"
- Line 150: "paramet[e]rized"

---

### Official Review · Reviewer_iRZ4 · 2025-03-19
**The authors propose an agent that efficiently explores its environment in the absence of a reward function based on the optimal explorer hypothesis, i.e. agents without explicit objectives. In particular, they focus on modeling the problem as a combinatorial optimization problem and exploring what the outcomes might be, both qualitatively and quantitatively. The problem is very interesting, and the underlying hypothesis is suggestive. As this is an experimental proposal, there are some gaps in the methodology for analyzing the results of the experiments. However, the general idea is very attractive since it could have important implications in the field, provided that the complexity analysis and other experimental elements are duly justified.**

**Rating:** 3
**Confidence:** 3

**Review:**

The authors propose an agent that efficiently explores its environment in the absence of a reward function based on the optimal explorer hypothesis, i.e. agents without explicit objectives. In particular, they focus on modeling the problem as a combinatorial optimization problem and exploring what the outcomes might be, both qualitatively and quantitatively. The problem is very interesting, and the underlying hypothesis is suggestive.

Comments

When modeling the problem as a combinatorial optimization problem, its complexity must be taken into account; the greater the number of variables, the greater the computational complexity.

Furthermore, given that the problem is presented as experimental, it would be important to mention the aspects that would be taken as parameters, i.e., the environment used, the number of variables, among others.

Furthermore, since the objective function is dynamic and dependent on uncertainty relationships, the sensitivity of the initial conditions should be explored.

---

### Decision · Program_Chairs · 2025-03-19

Accept